# Universal Dependencies Parsing for Colloquial Singaporean English

## Abstract

Singlish can be interesting to the ACL community both linguistically as a major creole based on English, and computationally for information extraction and sentiment analysis of regional social media. We investigate dependency parsing of Singlish by constructing a dependency treebank under the Universal Dependencies scheme, and then training a neural network model by integrating English syntactic knowledge into a state-of-the-art parser trained on the Singlish treebank. Results show that English knowledge can lead to 36% relative error reduction, resulting in a parser of around 85% accuracies. We make both our annotation and parser available for further research.

## 1 Introduction

Languages evolve temporally and geographically, both in vocabulary as well as in syntactic structures. When major languages such as English or French are adopted in another culture as the primary language, they often mix with existing languages or dialects in that culture and evolve into a stable language called a creole. Examples of creoles include the French-based Haitian Creole, and Colloquial Singaporean English (Singlish) (Mian-Lian and Platt, 1993), an English-based creole. While the majority of the natural language processing (NLP) research attention has been focused on the major languages, little work has been done on adapting the components to creoles. One notable body of work originated from the featured translation task of the EMNLP 2011 Workshop on Statistical Machine Translation (WMT11) to translate Haitian Creole SMS messages sent during the 2010 Haitian earthquake. This work highlights the importance of NLP tools on creoles in crisis situations for emergency relief (Hu et al., 2011; Hewavitharana et al., 2011).

Singlish is one of the major languages in Singapore, with borrowed vocabulary and grammars from a number of languages including Malay, Tamil, and Chinese dialects such as Hokkien, Cantonese and Teochew (Leimgruber, 2009, 2011), and it has been increasingly used in written forms on web media. Fluent English speakers unfamiliar with Singlish would find the creole hard to comprehend (Harada, 2009). Correspondingly, fundamental English NLP components such as POS taggers and dependency parsers perform poorly on such Singlish texts based on our observations. For example, Seah et al. (2015) adapted the Socher et al. (2013) sentiment analysis engine to the Singlish vocabulary, but failed to adapt the parser. Since dependency parsers are important for tasks such as information extraction (Miwa and Bansal, 2016) and discourse parsing (Li et al., 2015), this hinders the development of such downstream applications for Singlish in written forms and thus makes it crucial to build a dependency parser that can perform well natively on Singlish.

To address this issue, we start with investigating the linguistic characteristics of Singlish and specifically the causes of difficulties for understanding Singlish with English syntax. We found that, despite the obvious attribute of inheriting a large portion of basic vocabularies and grammars from English, Singlish not only imports terms from regional languages and dialects, its lexical semantics and syntax also deviate significantly from English (Leimgruber, 2009, 2011). We categorize the challenges and formalize their interpretation using Universal Dependencies (Nivre et al., 2016), which extends to the creation of a Singlish dependency treebank with 1,200 sentences.

Based on the intricate relationship between

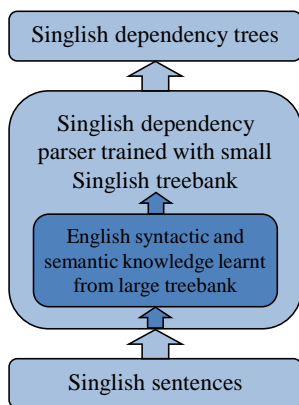

Figure 1: Overall model diagram

Singlish and English, we build a Singlish parser by leveraging knowledge of English syntax as a basis. This overall approach is illustrated in Figure 1. In particular, we train a basic Singlish parser with the best off-the-shelf neural dependency parsing model using biaffine attention (Dozat and Manning, 2016), and improve it with knowledge transfer by adopting neural stacking (Chen et al., 2016; Zhang and Weiss, 2016) to integrate the English syntax. Since POS tags are important features for dependency parsing (Chen and Manning, 2014; Dyer et al., 2015), we train a POS tagger for Singlish following the same idea by integrating English POS knowledge using neural stacking (Chen et al., 2016; Zhang and Weiss, 2016).

Results show that English syntax knowledge brings 31.58% and 36.51% relative error reduction on POS tagging and dependency parsing respectively, resulting in a Singlish dependency parser with 84.21% unlabeled attachment score (UAS).

We make our Singlish dependency treebank, the source code for training a dependency parser and the trained model for the parser with the best performance freely available online[1].

## 2 Related Work

Neural networks have led to significant advance in the performance for dependency parsing, including transition-based parsing (Chen and Manning, 2014; Zhou et al., 2015; Weiss et al., 2015; Dyer et al., 2015; Ballesteros et al., 2015; Andor et al., 2016), and graph-based parsing (Kiperwasser and Goldberg, 2016; Dozat and Manning, 2016). In particular, the biaffine attention method of Dozat and Manning (2016) uses deep bi-directional long

---

[1] https://github.com/ANONYMIZED/ANONYMIZED

short-term memory (bi-LSTM) networks for high-order non-linear feature extraction, producing the state-of-the-art performance for English dependency parsing. We adopt this model as the basis for our Singlish parser.

Our work belongs to a line of work on transfer learning for parsing, which leverages English resources in Universal Dependencies to improve the parsing accuracies of low-resource languages (Hwa et al., 2005; Cohen and Smith, 2009; Ganchev et al., 2009). Seminal work employed statistical models. McDonald et al. (2011) investigated delexicalized transfer, where word-based features are removed from a statistical model for English, so that POS and dependency label knowledge can be utilized for training a model for low-resource language. Subsequent work considered syntactic similarities between languages for better feature transfer (Täckström et al., 2012; Naseem et al., 2012; Zhang and Barzilay, 2015).

Recently, a line of work leverages neural network models for multi-lingual parsing (Guo et al., 2015; Duong et al., 2015; Ammar et al., 2016). The basic idea is to map the word embedding spaces between different languages into the same vector space, by using sentence-aligned bilingual data. This gives consistency in tokens, POS and dependency labels thanks to the availability of Universal Dependencies (Nivre et al., 2016). Our work is similar to these methods in using a neural network model for knowledge sharing between different languages. However, ours is different in the use of a neural stacking model, which respects the distributional differences between Singlish and English words. This empirically gives higher accuracies for Singlish.

## 3 Singlish Dependency Treebank

### 3.1 Universal Dependencies for Singlish

Since English is the major genesis of Singlish, we choose English as the source of lexical feature transfer to assist Singlish dependency parsing. Universal Dependencies provides a set of multilingual treebanks with cross-lingually consistent dependency-based lexicalist annotations, designed to aid development and evaluation for cross-lingual systems, such as multilingual parsers (Nivre et al., 2016). The current version of Universal Dependencies comprises not only major treebanks for 47 languages but also their siblings for domain-specific corpora and di-

alects. With the aligned initiatives for creating transfer-learning-friendly treebanks, we adopt the Universal Dependencies protocol for constructing the Singlish dependency treebank, both as a new resource for the low-resource languages and to facilitate knowledge transfer from English.

On top of the general Universal Dependencies guidelines, English-specific dependency relation definitions including additional subtypes are employed as the default standards for annotating the Singlish dependency treebank, unless augmented or redefined when necessary. The latest English corpus in Universal Dependencies v1.4[2] collection is constructed from the English Web Treebank (Bies et al., 2012), comprising of web media texts, which potentially smooths the knowledge transfer to our target Singlish texts in similar domains. The statistics of this dataset, from which we obtain English syntactic knowledge, is shown in Table 1 and we refer to this corpus as UD-Eng. This corpus uses 47 dependency relations and we show below how to conform to the same standard while adapting to unique Singlish grammars.

## 3.2 Challenges and Solutions for Annotating Singlish

The deviations of Singlish from English come from both the lexical and the grammatical levels (Leimgruber, 2009, 2011), which bring challenges for analysis on Singlish using English NLP tools. The former involves imported vocabularies from the first languages of the local people and the latter can be represented by a set of relatively localized features which collectively form 5 unique grammars of Singlish according to Leimgruber (2011). We find empirically that all these deviations can be accommodated by applying the existing English dependency relation definitions, which are explained with examples as follows.

**Imported vocabulary**: Singlish borrows a number of words and expressions from its non-English origins (Leimgruber, 2009, 2011), such as "Kiasu", which originates from Hokkien meaning "very anxious not to miss an opportunity".[3] These imported terms often constitute out-of-vocabulary (OOV) words with respect to a standard English treebank and result in difficulties for using English-trained tools on Singlish. All borrowed words are annotated according to their original

---

[2]Only guidelines for Universal Dependencies v2 but not the English corpus is available when this work is completed.

[3]Definition by the Oxford living Dictionaries for English.

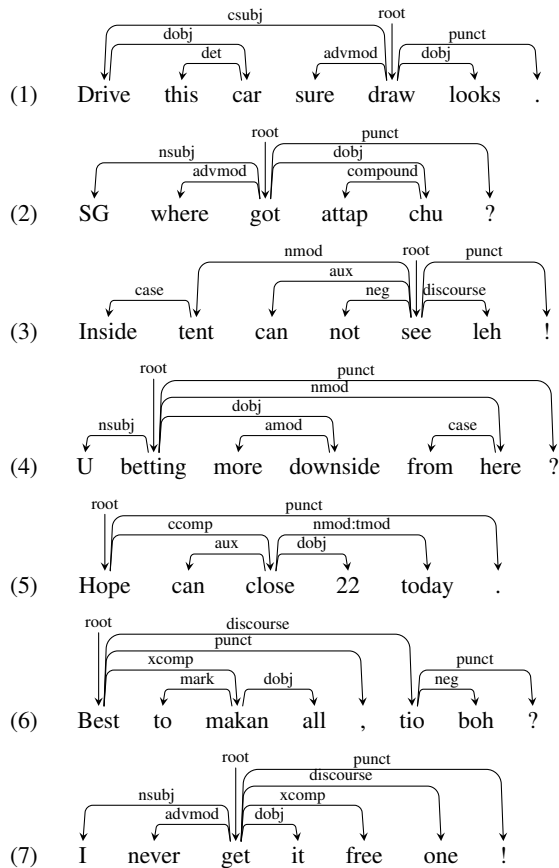

Figure 2: Unique Singlish grammars.

meanings using existing UD-Eng POS tags and dependency relations. Table A3 in Appendix A summarizes all borrowed terms in our treebank.

**Topic-prominence**: This type of sentences start with establishing its topic, which often serves as the default one that the rest of the sentence refers to, and they typically employ an object-subject-verb sentence structure (Leimgruber, 2009, 2011). In particular, three subtypes of topic-prominence are observed in the Singlish dependency treebank and their annotations are addressed as follows:

First, topics framed as clausal arguments at the beginning of the sentence are labeled as "csubj" (clausal subject), as shown by "Drive this car" of (1) in Figure 2. This type can be regarded as a variant of the English *it-extraposition* sentence structure with the extraposition moved to the front and the clause marker removed.

Second, noun phrases used to modify the predicate with the presence of a preposition is regarded as a "nsubj" (nominal subject). This is a common order of words used in Chinese and one example is the "SG" of (2) in Figure 2.

Third, prepositional phrases moved in front are

still treated as "nmod" (nominal modifier) of their intended heads, following the exact definition but as a Singlish-specific form of exemplification, as shown by the "Inside tent" of (3) in Figure 2.

**Copula deletion**: Imported from the corresponding Chinese sentence structure, this copula verb is often optional and deleted in Singlish, which is one of its diagnostic characteristics (Leimgruber, 2009, 2011). In UD-Eng standards, predicative "be" is the only verb used as a copula, which often depends on its complement to avoid copular head. Thus the deleted copula and its "cop" (copula) arc are simply ignored to preserve the intactness of the dependency tree, as shown by (4) in Figure 2. The only possibility of a copular root is when the copula has a clausal argument or adjunct. However, such copula is typically not deleted in either Singlish or Chinese, and we have not encountered such cases in our treebank. If they appear by any chance, feasible solutions can be either recovering the copula with a special symbol or promoting the head of the clausal argument or adjunct as the sentence root.

**NP deletion**: Noun-phrase (NP) deletion often results in null subjects or objects. It may be regarded as a branch of "Topic-prominence" but is a distinctive feature of Singlish with relatively high frequency of usage (Leimgruber, 2011). Again, we do not recover such relations since the deleted NP imposes negligible alteration to the dependency tree, as exemplified by (5) in Figure 2.

**Inversion**: Inversion in Singlish involves either reversed order of subject and verb in interrogative sentences, or tag questions in polar interrogatives (Leimgruber, 2011). The former simply involves a change of word orders and thus requires no special treatments. On the other hand, tag questions should be carefully analyzed in two scenarios. One type is in the form of "isn't it ?" or "haven't you ?", which are dependents of the sentence root with the "parataxis" relation.[4] The other type is exemplified as "right ?", and its Singlish equivalent "tio boh ?" (a transliteration from Hokkien) are labeled with the "discourse" (discourse element) relation with respect to the sentence root. See example (6) in Figure 2.

**Discourse particles**: Usage of clausal-final discourse particles, which originates from Hokkien

and Cantonese, is one of the most typical feature of Singlish (Leimgruber, 2009, 2011; Lim, 2007). All discourse particles that appear in our treebank are summarized in Table A3 in Appendix A with the imported vocabulary:. These words express the tone of the sentence and thus have the "INTJ" (interjection) POS tag and depend on the root of the sentence or clause labeled with "discourse", as is shown by the "leh" of (3) in Figure 2. The word "one" is a special instance of this type with the sole purpose being a tone marker in Singlish but not English, as shown by (7) in Figure 2.

## 3.3 Data Selection and Annotation

**Data Source**: Singlish is used in written form mainly in social media and local Internet forums. After comparison, we chose the *SG Talk Forum*[5] as our data source due to its relative abundance in Singlish contents. We crawled 84,459 posts using the Scrapy framework[6] from pages dated up to 25th December 2016, retaining sentences of length between 5 and 50, which total 58,310. Sentences are reversely sorted according to the log likelihood of the sentence given by an English language model trained using the KenLM toolkit (Heafield et al., 2013)[7] normalized by the sentence length, so that those most different from standard English can be chosen. Among the top 10,000 sentences, 1,977 sentences contain unique Singlish vocabularies defined by *The Coxford Singlish Dictionary*[8], *A Dictionary of Singlish and Singapore English*[9], and the *Singlish Vocabulary* Wikipedia page[10]. The average normalized log likelihood of these 10,000 sentences is -5.81, and the same measure for all sentences in UD-Eng is -4.81. This means these sentences with Singlish contents are 10 times less probable expressed as standard English than the UD-Eng contents in the web domain. This contrast indicates the degree of lexical deviation of Singlish from English. We chose 1,200 sentences from the first 10,000.

**Annotation**: The chosen texts are divided by random selection into training, development, and testing sets according to the proportion of sentences in the training, development, and test di-

---

[4]Relation between the main verb of a clause and other sentential elements, such as sentential parenthetical clause, or adjacent sentences without any explicit coordination or subordination.

[5]http://sgTalk.com
[6]https://scrapy.org/
[7]Trained using the *afp_eng* and *xin_eng* sources of English Gigaword Fifth Edition (Gigaword).
[8]http://72.5.72.93/html/lexec.php
[9]http://www.singlishdictionary.com
[10]https://en.wikipedia.org/wiki/Singlish_vocabulary

|  | UD English | | Singlish | |
|---|---|---|---|---|
|  | Sentences | Words | Sentences | Words |
| Train | 12,543 | 204,586 | 900 | 8,221 |
| Dev | 2,002 | 25,148 | 150 | 1,384 |
| Test | 2,077 | 25,096 | 150 | 1,381 |

Table 1: Division of training, development, and test sets for Singlish Treebank

vision for UD-Eng, as summarized in Table 1. The sentences are tokenized using the NLTK Tokenizer,[11] and then annotated using the Dependency Viewer.[12] In total, all 17 UD-Eng POS tags and 41 out of the 47 UD-Eng dependency labels are present in the Singlish dependency treebank. Besides, 100 sentences are randomly selected and double annotated by one of the coauthors, and the inter-annotator agreement has an unlabeled attachment score (UAS) of 85.30% and a labeled attachment score (LAS) of 75.72%. A full summary of the number of occurences of each POS tag and dependency labels is included in Appendix A.

## 4 Part-of-Speech Tagging

In order to obtain automatically predicted POS tags as features for a base English dependency parser, we train a POS tagger for UD-Eng using the baseline model of Chen et al. (2016), depicted in Figure 3. The bi-LSTM networks with a CRF layer (bi-LSTM-CRF) have shown state-of-the-art performance by globally optimizing the tag sequence (Huang et al., 2015; Chen et al., 2016). Based on this English POS tagging model, we train a POS tagger for Singlish using the feature-level neural stacking model of Chen et al. (2016). Both the English and Singlish models consist of an input layer, a feature layer, and an output layer.

### 4.1 Base Bi-LSTM-CRF POS Tagger

**Input Layer**: Each token is represented as a vector by concatenating a word embedding from a lookup table with a weighted average of its character embeddings given by the attention model of Bahdanau et al. (2014). Following Chen et al. (2016), the input layer produces a dense representation for the current input token by concatenating its word vector and the ones for its surrounding context tokens in a window of finite size.

---

[11]http://www.nltk.org/api/nltk.tokenize.html

[12]http://nlp.nju.edu.cn/tanggc/tools/DependencyViewer.exe

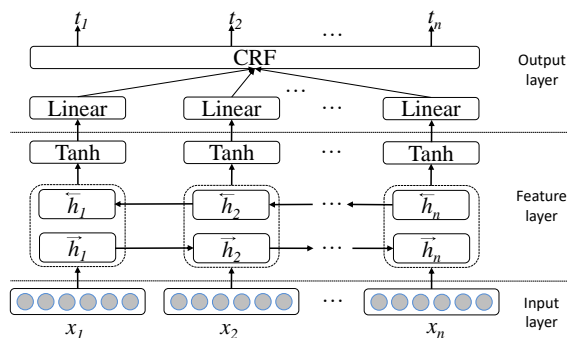

Figure 3: Base POS tagger

**Feature Layer**: This layer employs a bi-LSTM network to encode the input into a sequence of hidden vectors that embody global contextual information. Following Chen et al. (2016), we adopt bi-LSTM with peephole connections (Graves and Schmidhuber, 2005).

**Output layer**: This is a CRF layer to predict the POS tags for the input words by maximizing the conditional probability of the sequence of tags given input sentence.

### 4.2 POS Tagger with Neural Stacking

We adopt the deep integration neural stacking structure presented in Chen et al. (2016). As shown in Figure 4, the distributed vector representation for the target word at the input layer of the Singlish Tagger is augmented by concatenating the emission vector produced by the English Tagger with the original word and character-based embeddings, before applying the concatenation within a context window in section 4.1. During training, loss is back-propagated to all trainable parameters in both the Singlish Tagger and the pre-trained feature layer of the base English Tagger. At test time, the input sentence is fed to the integrated tagger model as a whole for inference.

### 4.3 Results

We use the publicly available source code[13] by Chen et al. (2016) to train a 1-layer bi-LSTM-CRF based POS tagger on UD-Eng, using 50-dimension pre-trained SENNA word embeddings (Collobert et al., 2011). We set the hidden layer size to 300, the initial learning rate for Adagrad (Duchi et al., 2011) to 0.01, the regularization parameter $\lambda$ to $10^{-6}$, and the dropout rate to 15%. The tagger gives 94.84% accuracy on the UD-Eng

---

[13]https://github.com/chenhongshen/NNHetSeq

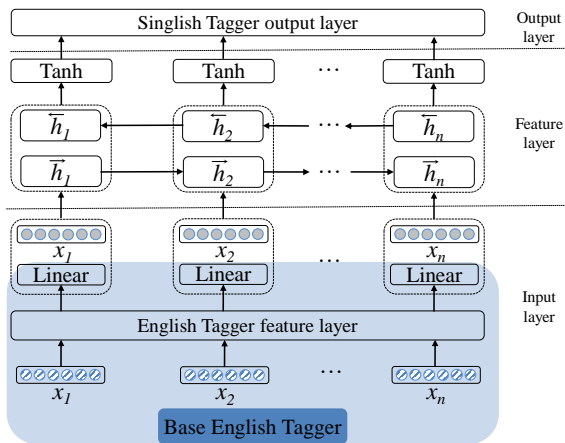

Figure 4: POS tagger with neural stacking

| System | Accuracy |
|---|---|
| ENG-on-SIN | 80.67% |
| Base-ICE-SIN | 77.77% |
| Stack-ICE-SIN | **84.79%** |

Table 2: POS tagging accuracies

test set after 24 epochs, chosen according to development tests, which is comparable to the state-of-the-art accuracy of 95.17% reported by Plank et al. (2016). We use these settings to perform 10-fold jackknifing of POS tagging on the UD-Eng training set, with an average accuracy of 95.60%.

Similarly, we trained a POS tagger using the Singlish dependency treebank alone with pre-trained word embeddings on The Singapore Component of the International Corpus of English (ICE-SIN) (Nihilani, 1992; Ooi, 1997), which consists of both spoken and written texts. However, due to limited amount of training data, the tagging accuracy is not satisfactory even with a large dropout rate to avoid overfitting. In contrast, the neural stacking structure on top of the English base model trained on UD-Eng achieves a POS tagging accuracy of 84.79%[14], which corresponds to a 31.58% relative error reduction over the baseline Singlish model, as shown in Table 2. We use this for 10-fold jackknifing on Singlish parsing training data, and tagging the Singlish development and test data.

---

[14]We empirically find that using ICE-SIN embeddings in neural stacking model performs better than using English SENNA embeddings. Similar findings are found for the parser, of which more details are given in section 6.

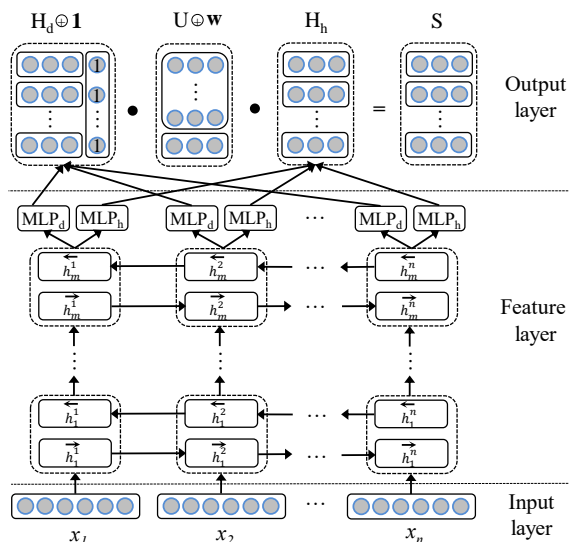

Figure 5: Base parser

## 5 Dependency Parsing

We adopt the Dozat and Manning (2016) parser[15] as our base model, as displayed in Figure 5, and apply neural stacking to achieve improvements over the baseline parser. Both the base and neural stacking models consist of an input layer, a feature layer, and an output layer.

### 5.1 Base Parser with Bi-affine Attentions

**Input Layer**: This layer encodes the current input word by concatenating a pre-trained word embedding with a trainable word embedding and POS tag embedding from the respective lookup tables.

**Feature Layer**: The two recurrent vectors produced by the multi-layer bi-LSTM network from each input vector are concatenated and mapped to multiple feature vectors in lower-dimension space by a set of parallel multilayer perceptron (MLP) layers. Following Dozat and Manning (2016), we adopt Cif-LSTM cells (Greff et al., 2016).

**Output Layer**: This layer applies biaffine transformation on the feature vectors to calculate the score of the directed arcs between every pair of words. The inferred trees for input sentence are formed by choosing the head with the highest score for each word and a cross-entropy loss is calculated to update the model parameters.

### 5.2 Parser with Neural Stacking

Inspired by the idea of feature-level neural stacking (Chen et al., 2016; Zhang and Weiss, 2016), we concatenate the pre-trained word embedding,

---

[15]https://github.com/tdozat/Parser

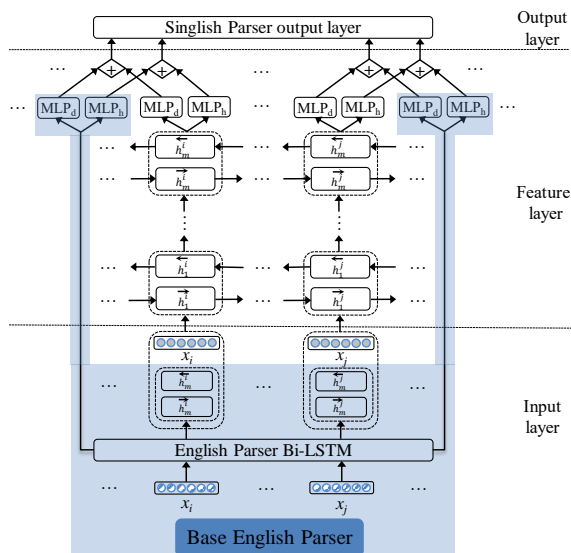

Figure 6: Parser with neural stacking

trainable word and tag embeddings, with the two recurrent state vectors at the last bi-LSTM layer of the English Tagger as the input vector for each target word. In order to further preserve syntactic knowledge retained by the English Tagger, the feature vectors from its MLP layer is added to the ones produced by the Singlish Parser, as illustrated in Figure 6, and the scoring tensor of the Singlish Parser is initialized with the one from the trained English Tagger. Loss is back-propagated by reversely traversing all forward paths to all trainable parameter for training and the whole model is used collectively for inference.

# 6 Experiments and Discussion

## 6.1 Experimental Settings

We train an English parser on UD-Eng with the default model settings in Dozat and Manning (2016). It achieves a UAS of 88.88% and a LAS of 85.00%, which are close to the state-of-the-art 85.90% LAS on UD-Eng reported by Ammar et al. (2016), and the main difference is caused by us not using fine-grained POS tags. We apply the same settings for a baseline Singlish parser. We attempt to choose a better configuration of the number of bi-LSTM layers and the hidden dimension based on the development set performance, but the default settings turn out to perform the best. Thus we stick to all default hyperparameters in Dozat and Manning (2016) for training the Singlish parsers.

We experimented with different word embeddings, as with the raw text sources summarized in Table 3 and further described in 6.2. When using

| | Sentences | Words | Vocabulary |
|---|---|---|---|
| GloVe6B | N.A. | 6000m | 400,000 |
| Giga100M | 57,000 | 1.26m | 54,554 |
| ICE-SIN | 87,084 | 1.26m | 40,532 |

Table 3: Comparison of the scale of sources for training word embeddings

| Trained on | System | UAS | LAS |
|---|---|---|---|
| English | ENG-on-SIN | 76.15 | 65.62 |
| Singlish | Baseline | 75.13 | 64.01 |
| | Base-Giga100M | 76.40 | 64.94 |
| | Base-GloVe6B | 78.01 | 68.85 |
| | Base-ICE-SIN | 79.63 | 68.00 |
| Both | Stack-ICE-SIN | **84.21** | **76.57** |

Table 4: Dependency parser performances

the neural stacking model, we fix the model configuration for the base English parser model and choose the size of the hidden vector and the number of bi-LSTM layers stacked on top based on the development set. It turns out that a 1-layer bi-LSTM with 750 hidden dimension performs the best, where the bigger hidden layer accommodates the elongated input vector to the stacked bi-LSTM and the fewer number of recurrent layers avoids overfitting on the small Singlish dependency treebank, given the deep bi-LSTM English parser network at the bottom.

## 6.2 Investigating Distributed Lexical Characteristics

In order to learn characteristics of distributed lexical semantics for Singlish, we compare performances of the Singlish dependency parser using several sets of pre-rained word embeddings: GloVe6B, large-scale English word embeddings[16]; ICE-SIN, Singlish word embeddings trained using GloVe (Pennington et al., 2014) on the ICE-SIN (Nihilani, 1992; Ooi, 1997) corpus; Giga100M, a small-scale English word embeddings trained using GloVe (Pennington et al., 2014) with the same settings on a comparable size of English data randomly selected from the English Gigaword Fifth Edition for a fair comparison with ICE-SIN embeddings.

First, the English Giga100M embeddings marginally improve the Singlish parser from the baseline without pre-trained embeddings, giving a performance on par with using UD-Eng parser directly on Singlish, represented as "ENG-on-SIN"

---

[16]Trained with Wikipedia 2014 the Gigaword. Downloadable from http://nlp.stanford.edu/data/glove.6B.zip

| | Topic Prominence | | Copula Deletion | | NP Deletion | | Discourse Particles | | Others | |
|---|---|---|---|---|---|---|---|---|---|---|
| Sentences | 15 | | 19 | | 21 | | 51 | | 67 | |
| | UAS | LAS | UAS | LAS | UAS | LAS | UAS | LAS | UAS | LAS |
| ENG-on-SIN | 73.15 | 56.48 | 61.87 | 51.08 | 70.29 | 60.57 | 70.75 | 60.00 | 81.90 | 71.83 |
| Base-Giga100M | 77.78 | 62.96 | **79.86** | 66.91 | 76.00 | 65.14 | 83.75 | 74.75 | 70.34 | 58.40 |
| Base-ICE | 74.07 | 64.81 | 71.94 | 61.87 | **79.43** | 70.86 | 87.25 | 77.50 | 77.61 | 63.99 |
| Stack-ICE | **79.63** | **69.44** | **79.86** | **71.22** | 78.86 | **71.43** | **88.00** | **82.75** | **85.07** | **76.87** |

Table 5: Error analysis with respect to grammar types

in Table 4. With much more English lexical semantics being fed to the Singlish parser using the English GloVe6B embeddings, the enhancement is still marginal. On the contrary, the Singlish ICE-SIN embeddings lead to more improvement, with 18.09% relative error reduction compared with 5.11% using the English Giga100M embeddings. It is even better than using the English GloVe6B embeddings despite the huge difference in sizes. This demonstrate the distributional differences between Singlish and English tokens, even though they share a large vocabulary. More detailed comparison is described in section 6.4.

### 6.3 Knowledge Transfer Using Neural Stacking

We train a parser with neural stacking and Singlish ICE-SIN embeddings, which achieves the best performance among all the models, with a UAS of 84.21% as shown in Table 4, which corresponds to 36.51% relative error reduction compared to the baseline. This demonstrates that knowledge from English can be successfully incorporated to boost the Singlish parser. This significant improvement is further explained below.

### 6.4 Improvements over Grammar Types

To analyze the sources of improvements for Singlish parsing using different model configurations, we conduct error analysis over 5 syntactic categories[17], including 4 types of grammars mentioned in section 3.2[18], and 1 for all other cases, including sentences containing imported vocabularies but expressed in basic English syntax. The number of sentences and the results in each group of the test set are shown in Table 5.

The neural stacking model leads to the biggest improvement over nearly all categories except for a slightly lower yet competitive performance on "NP Deletion" cases, which explains the significant overall improvement.

Comparing the base model with ICE-SIN embeddings with the base parser trained on UD-Eng, which contain syntactic and semantic knowledge in Singlish and English, respectively, the former outperforms the latter on all 4 types of Singlish grammars but not for the remaining samples. This suggests that the base English parser mainly contributes to analyzing basic English syntax, while the Singlish dependency treebank models unique Singlish grammars better.

Using the English Giga100M embeddings in the base model only helps to improve parsing for "Topic Prominence" and "Copula Deletion" compared to the Singlish ICE-SIN embeddings. The main reason can be that these two types of grammars preserve a high percentage of English syntax, with only alterations in word orders and deletions of one word. However, "NP Deletion" results in more radical sentence incompleteness and "Discourse Particles" are unique in Singlish. An interesting finding is that feeding English distributed lexical semantic information to the base Singlish parser undermines the performance even over basic English syntax, which again suggests the differences in distributed lexical semantics.

## 7 Conclusion

We have investigated dependency parsing for Singlish, an important English-based creole language, through annotations of a Singlish dependency treebank with 10,986 words, and building an enhanced parser by leveraging on knowledge transferred from a 20-times-bigger English treebank of Universal Dependencies. We demonstrate the effectiveness of using neural stacking for feature transfer by boosting the Singlish dependency parsing performance to 84.21% UAS, with a 36.51% error reduction. We release the annotated Singlish dependency treebank, the trained model and the source code for the parser with free public access. Possible future work include expanding the investigation to other regional languages such as Malay and Indonesian.

---

[17]Multiple labels are allowed for one sentence.

[18]The "Inversion" type of grammar is not analyzed since there is only 1 such sentence in the test set.

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

# A  Statistics of Singlish Dependency Treebank

| POS Tags | | | | | |
|---|---|---|---|---|---|
| ADJ | 767 | INTJ | 510 | PUNCT | 623 |
| ADP | 487 | NOUN | 1717 | SCONJ | 126 |
| ADV | 912 | NUM | 147 | SYM | 11 |
| AUX | 429 | PART | 354 | VERB | 1678 |
| CONJ | 167 | PRON | 675 | X | 10 |
| DET | 386 | PROPN | 787 | | |

Table A1: Statistics of POS tags

| Dependency labels | | | |
|---|---|---|---|
| acl | 33 | dobj | 590 |
| acl:relcl | 29 | expl | 10 |
| advcl | 186 | iobj | 15 |
| advmod | 829 | list | 10 |
| appos | 18 | mwe | 100 |
| amod | 422 | name | 117 |
| aux | 377 | neg | 260 |
| auxpass | 47 | nmod | 364 |
| case | 463 | nmod:npmod | 26 |
| cc | 167 | nmod:poss | 153 |
| ccomp | 135 | nmod:tmod | 77 |
| compound | 420 | nsubj | 1005 |
| compound:prt | 27 | nsubjpass | 34 |
| conj | 229 | nummod | 94 |
| cop | 152 | mark | 275 |
| csubj | 30 | parataxis | 223 |
| det | 304 | punct | 626 |
| det:predet | 7 | remnant | 16 |
| discourse | 507 | vocative | 38 |
| dislocated | 2 | xcomp | 187 |

Table A2: Statistics of dependency labels

| A-B | | |
|---|---|---|
| act blur | ah | ah beng |
| ah ne | angpow | arrowed |
| aiyah | ang ku kueh | angmoh/ang moh |
| ahpek / ah peks | | atas |
| ba | boh/bo | boho jiak |
| boh pian | buay lin chu | buen kuey |

| C | | |
|---|---|---|
| chai tow kway | chao ah beng | chap chye png |
| char kway teow | chee cheong fun / | che cheong fen |
| cheesepie | cheong / chiong | chiam / cham |
| chiak liao bee / jiao liao bee | | chio |
| ching chong | chio bu / chiobu | chui |
| chop chop | chow-angmoh | chwee kueh |

| D-F | | |
|---|---|---|
| dey | diam diam | die kock standing |
| die pain pain | dun | eat grass |
| flip prata | fried beehoon | |

| G | | |
|---|---|---|
| gahmen / garment | gam | geylang |
| gone case | gong kia | goreng pisang |
| gui | | |

| H-J | | |
|---|---|---|
| hah / har / huh | hai si lang | heng |
| hiak hiak hiak | hiong | hoot |
| hor | Hosay / ho say | how lian |
| huat | jepun kia / jepun kias | |
| jialat / jia lak / jia lat | | |

| K | | |
|---|---|---|
| ka | kaki kong kaki song | |
| kancheong | kateks | kautim |
| kay kiang | kayu | kee chia |
| kee siao | kelong | kena / kana |
| kiam | kiasu | ki seow |
| kkj | kong si mi | kopi |
| kopi lui | kopi-o | kosong |
| koyok | ku ku bird | |

| L | | |
|---|---|---|
| lagi | lai liao | laksa |
| la / lah | lao jio kong | lao sai |
| lau chwee nua | lau | leh |
| liao / ler | like dat / like that | |
| lim peh | lobang | loh / lor |

| M | | |
|---|---|---|
| mahjong kaki | makan | ma / mah |
| masak masak | mati | mee |
| mee pok | mee rebus | mee siam |
| mee sua | mei mei | |

| N-S | | |
|---|---|---|
| nasi lemak | pang sai | piak |
| sabo | sai | same same |
| sia | sianz / sian | sia suay |
| sibeh | siew dai | siew siew dai |
| simi taisee | soon kuey | sotong |
| suay / suey | swee | |

| T | | |
|---|---|---|
| tahan | tak pakai | te te kee |
| talk cock / talk cock sing song | | tikopeh |
| tio | tio pian/dio pian | |
| tong | tua | |

| U-Z | | |
|---|---|---|
| umm zai | up lorry / up one's lorry | |
| wahlow / wah lau | walaneh / wah lan eh | |
| wa / wah | xiao | ya ya |
| zhun / buay zhun | | |

Table A3: Imported vocabulary and discourse particles in the Singlish dependency treebank

