# Peer review of "Universal Dependencies Parsing for Colloquial Singaporean English"

_ACL 2017 — decision unknown_

[Official Review · Reviewer 1 · rating 4 · confidence 4]
soundness 5 · originality 5 · clarity 5 · impact 3 · substance 4 · appropriateness 5 · meaningful comparison 3 · presentation format Poster

The paper describes a deep-learning-based model for parsing the creole
Singaporean English to Universal Dependencies. They implement a parser based on
the model by Dozat and Manning (2016) and add neural stacking (Chen et al.,
2016) to it. They train an English model and then use some of the hidden
representations of the English model as input to their Singlish parser. This
allows them to make use of the much larger English training set along with a
small Singlish treebank, which they annotate. They show that their approach
(LAS 76.57) works better than just using an English parser (LAS 65.6) or
training a parser on their small Singlish data set (LAS 64.01). They also
analyze for which
common constructions, their approach improves parsing quality. 

They also describe and evaluate a stacked POS model based on Chen et al.
(2016), they discuss how common constructions should be analyzed in the UD
framework, and they provide an annotated treebank of 1,200 sentences. 100 of
them were annotated by two people and their inter-annotator agreement was 85.3
UAS and 75.7 LAS.

- Strengths:

 - They obtain good results and their experimental setup appears to be solid.

 - They perform many careful analyses and explore the influence on many
parameters of their model.

 - They provide a small Singlish treebank annotated according to the Universal
Dependencies v1.4 guidelines.

 - They propose very sound guidelines on how to analyze common Singlish
constructions in UD.

 - Their method is linguistically informed and they nicely exploit similarity
between standard English and the creole Singaporean English.

 - The paper presents methods for a low-resource language.

 - They are not just applying an existing English method to another language
but instead present a method that can be potentially used for other closely
related language pairs.

 - They use a well-motivated method for selecting the sentences to include in
their treebank.

 - The paper is very well written and easy to read.

- Weaknesses:

 - The annotation quality seems to be rather poor. They performed double
annotation of 100 sentences and their inter-annotator agreement is just 75.72%
in terms of LAS. This makes it hard to assess how reliable the estimate of the
LAS of their model is, and the LAS of their model is in fact slightly higher
than the inter-annotator agreement. 

UPDATE: Their rebuttal convincingly argued that the second annotator who just
annotated the 100 examples to compute the IAA didn't follow the annotation
guidelines for several common constructions. Once the second annotator fixed
these issues, the IAA was reasonable, so I no longer consider this a real
issue.

- General Discussion:

I am a bit concerned about the apparently rather poor annotation quality of the
data and how this might influence the results, but overall, I liked the paper
a lot and I think this would be a good contribution to the conference.

- Questions for the authors:

 - Who annotated the sentences? You just mention that 100 sentences were
annotated by one of the authors to compute inter=annotator agreement but you
don't mention who annotated all the sentences.

 - Why was the inter-annotator agreement so low? In which cases was there
disagreement? Did you subsequently discuss and fix the sentences for which
there was disagreement?

 - Table A2: There seem to be a lot of discourse relations (almost as many as
dobj relations) in your treebank. Is this just an artifact of the colloquial
language or did you use "discourse" for things that are not considered
"discourse" in other languages in UD?

 - Table A3: Are all of these discourse particles or discourse + imported
vocab? If the latter, perhaps put them in separate tables, and glosses would be
helpful.

- Low-level comments:

 - It would have been interesting if you had compared your approach to the one
by Martinez et al. (2017, https://arxiv.org/pdf/1701.03163.pdf). Perhaps you
should mention this paper in the reference section.

 - You use the word "grammar" in a slightly strange way. I think replacing
"grammar" with syntactic constructions would make it clearer what you try to
convey. (e.g., line 90)

 - Line 291: I don't think this can be regarded as a variant of
it-extraposition. But I agree with the analysis in Figure 2, so perhaps just
get rid of this sentence.

 - Line 152: I think the model by Dozat and Manning (2016) is no longer
state-of-the art, so perhaps just replace it with "very high performing model"
or something like that.

 - It would be helpful if you provided glosses in Figure 2.

[Official Review · Reviewer 2 · rating 4 · confidence 5]
soundness 5 · originality 5 · clarity 5 · impact 3 · substance 3 · appropriateness 5 · meaningful comparison 3 · presentation format Poster

- Strengths:
Nice results, nice data set. Not so much work on Creole-like languages,
especially English.  

- Weaknesses:
A global feeling of "Deja-vu", a lot of similar techniques have been applied to
other domains, other ressource-low languages. Replace word embeddings by
clusters and neural models by whatever was in fashion 5 years ago and we can
find more or less the same applied to Urdu or out-of-domain parsing. I liked
this paper though, but I would have appreciated the authors to highlight more
their contributions and position their work better within the literature.

- General Discussion:

This paper presents a set of experiments designed a) to show the effectiveness
of a neural parser  in a scarce resource scenario and b) to introduce a new
data set of Creole English (from Singapour, called Singlish). While this data
set is relatively small (1200 annotated sentences, used with 80k unlabeled
sentences for word embeddings induction), the authors manage to present
respectable results via interesting approach even though using features from
relatively close languages are not unknown from the parsing community (see all
the line of work on parsing Urdu/Hindi, on Arabic dialect using MSA based
parsers, and so on).
Assuming we can see Singlish as an extreme of Out-of-domain English and given
all the set of experiments, I wonder why the authors didn’t try the classical
technique on domain-adaptation, namely training with UD_EN+90% of the Singlish
within a 10 cross fold experiment ? just so we can have another interesting
baseline (with and without word embeddings, with bi-lingual embeddings if
enough parallel data is available).
I think that paper is interesting but I really would have appreciated more
positioning regarding all previous work in parsing low-ressources languages and
extreme domain adaptation. A table presenting some results for Irish and other
very small treebanks would be nice.
Also how come the IAA is so low regarding the labeled relations?

*****************************************
Note after reading the authors' answer
*****************************************

Thanks for your clarifications (especially for redoing the IAA evaluation). I
raised my recommendation to 4, I hope it'll get accepted.

[Official Review · Reviewer 3 · rating 4 · confidence 3]
soundness 5 · originality 5 · clarity 5 · impact 3 · substance 4 · appropriateness 5 · meaningful comparison 3 · presentation format Poster

The authors construct a new dataset of 1200 Singaporean English (Singlish)
sentences annotated with Universal Dependencies. They show that they can
improve the performance of a POS tagger and a dependency parser on the Singlish
corpus by integrating English syntactic knowledge via a neural stacking model.

- Strengths:
Singlish is a low-resource language. The NLP community needs more data for low
resource languages, and the dataset accompanying this paper is a useful
contribution. There is also relatively little NLP research on creoles, and the
potential of using transfer-learning to analyze creoles, and this paper makes a
nice contribution in that area.

The experimental setup used by the authors is clear. They provide convincing
evidence that incorporating knowledge from an English-trained parser into a
Singlish parser outperforms both an English-only parser and a Singlish-only
parser on the Singlish data. They also provide a good overview of the relevant
differences between English and Singlish for the purposes of syntactic parser
and a useful analysis of how different parsing models handle these
Singlish-specific constructions.

- Weaknesses:

There are three main issues I see with this paper:
*  There is insufficient comparison to the UD annotation of non-English
languages. Many of the constructions they bring up as specific to Singlish are
also present in other UD languages, and the annotations should ideally be
consistent between Singlish and these languages.
*  I'd like to see an analysis on the impact of training data size. A central
claim of this paper is that using English data can improve performance on a
low-resource language like Singlish. How much more Singlish data would be
needed before the English data became unnecessary?
*  What happens if you train a single POS/dep parsing model on the concatenated
UD Web and Singlish datasets? This is much simpler than incorporating neural
stacking. The case for neural stacking is stronger if it can outperform this
baseline.

- General Discussion:
Line 073: “POS taggers and dependency parsers perform poorly on such Singlish
texts based on our observations” - be more clear that you will quantify this
later. As such, it seems a bit hand-wavy.

Line 169: Comparison to neural network models for multi-lingual parsing. As far
as I can tell, you don't directly try the approach of mapping Singlish and
English word embeddings into the same embedding space.

Line 212: Introduction of UD Eng. At this point, it is appropriate to point out
that the Singlish data is also web data, so the domain matches UD Eng.

Line 245: “All borrowed words are annotated according to their original
meanings”. Does this mean they have the same POS as in  the language from
which they were borrowed? Or the POS of their usage in Singlish?

Figure 2: Standard English glosses would be very useful in understanding the
constructions and checking the correctness of the UD relations used.

Line 280: Topic prominence: You should compare with the “dislocated” label
in UD. From the UD paper: “The dislocated relation captures preposed (topics)
and postposed elements”. The syntax you are describing sounds similar to a
topic-comment-style syntax; if it is different, then you should make it clear
how.

Line 294: “Second, noun phrases used to modify the predicate with the
presence of a preposition is regarded as a “nsubj” (nominal subject).”
Here, I need a gloss to determine if this analysis makes sense. If the phrase
is really being used to modify the predicate, then this should not be nsubj. UD
makes a distinction between core arguments (nsubj, dobj, etc) and modifiers. If
this is a case of modification, then you should use one of the modification
relations, not a core argument relation. Should clarify the language here.

Line 308: “In UD-Eng standards, predicative “be” is the only verb used as
a copula, which often depends on its complement to avoid copular head.” This
is an explicit decision made in UD, to increase parallelism with non-copular
languages (e.g., Singlish). You should call this out. I think the rest of the
discussion of copula handling is not necessary.

Line 322: “NP deletion: Noun-phrase (NP) deletion often results in null
subjects or objects.” This is common in other languages (zero-anaphora in
e.g. Spanish, Italian, Russian, Japanese… )Would be good to point this out,
and also point to how this is dealt with in UD in those languages (I believe
the same way you handle it).

Ling 330: Subj/verb inversion is common in interrogatives in other languages
(“Fue Marta al supermercado/Did Marta go to the supermarket?”). Tag
questions are present in English (though perhaps are not as frequent). You
should make sure that your analysis is consistent with these languages.

Sec 3.3 Data Selection and Annotation:
The way you chose the Singlish sentences, of course an English parser will do
poorly (they are chosen to be disimilar to sentences an English parser has seen
before). But do you have a sense of how a standard English parser does overall
on Singlish, if it is not filtered this way? How common are sentences with
out-of-vocabulary terms or the constructions you discussed in 3.2?

A language will not necessarily capture unusual sentence structure,
particularly around long-distance dependencies. Did you investigate whether
this method did a good job of capturing sentences with the grammatical
differences to English you discussed in Section 3.2?

Line 415: “the inter-annotator agreement has an unlabeled attachment score
(UAS) of 85.30% and a labeled attachment score (LAS) of 75.72%.”
*  What’s the agreement on POS tags? Is this integrated with LAS?
*  Note that in Silveira et al 2014, which produced UD-Eng, they measured 94%
inter-annotator agreement on a per-token basis. Why the discrepancy?

POS tagging and dep parsing sections:
For both POS-tagging and dep parsing, I’d like to see some analysis on the
effect of training set size. E.g., how much more Singlish data would be needed
to train a POS tagger/dep parser entirely on Singlish and get the same accuracy
as the stacked model?

What happens if you just concatenate the datasets? E.g., train a model on a
hybrid dataset of EN and Singlish, and see what the result is?

Line 681: typo: “pre-rained” should be “pre-trained”

742 “The neural stacking model leads to the biggest improvement over nearly
all categories except for a slightly lower yet competitive performance on “NP
Deletion” cases” --- seems that the English data strongly biases the parser
to expect an explicit subj/obj. you could try deleting subj/obj from some
English sentences to improve performance on this construction.